# LogicReward: Incentivizing LLM Reasoning via Step-Wise Logical Supervision

**Jundong Xu**[1]*, **Hao Fei**[1]†, **Huichi Zhou**[2], **Xin Quan**[3], **Qijun Huang**[4], **Shengqiong Wu**[1], **William Yang Wang**[5], **Mong-Li Lee**[1], **Wynne Hsu**[1]

[1]National University of Singapore, [2]University College London, [3]University of Manchester, [4]University of Melbourne, [5]University of California, Santa Barbara

## Abstract

Although LLMs exhibit strong reasoning capabilities, existing training methods largely depend on outcome-based feedback, which can produce correct answers with flawed reasoning. Prior work introduces supervision on intermediate steps but still lacks guarantees of logical soundness, which is crucial in high-stakes scenarios where logical consistency is paramount. To address this, we propose **LogicReward**, a novel reward function that guides model training by enforcing step-level logical correctness with a theorem prover. We further introduce *Autoformalization with Soft Unification*, which reduces natural language ambiguity and improves formalization quality, enabling more effective use of the theorem prover. An 8B model trained on data constructed with LogicReward surpasses GPT-4o and o4-mini by 11.6% and 2% on natural language inference and logical reasoning tasks with simple training procedures. Further analysis shows that LogicReward enhances reasoning faithfulness, improves generalizability to unseen tasks such as math and commonsense reasoning, and provides a reliable reward signal even without ground-truth labels. The code and data are available at `https://llm-symbol.github.io/LogicReward`.

## 1 Introduction

Reasoning is a pinnacle of human cognition, and enabling large language models (LLMs) to perform rigorous, human-like reasoning is a key step towards achieving artificial superintelligence (Feng et al., 2024; Li et al., 2025b). In prior work, a representative approach to improve LLM reasoning is to prompt the model to "think step by step", termed chain-of-thought (CoT) prompting, which has been shown to enhance LLMs' reasoning performance (Wei et al., 2022b). More recently, researchers have found that increasing a model's inference-time computation by allowing it to generate longer reasoning chains can further enhance its reasoning capabilities (Liang et al., 2024; Snell et al., 2024). Representative works, e.g., DeepSeek-R1 (Guo et al., 2025), adopt this paradigm by harnessing the answer as a feedback signal during training to incentivize longer and more thorough reasoning chains, which enable it to tackle more complex reasoning tasks. Nevertheless, relying solely on final-outcome feedback means the reasoning process receives no direct supervision. Consequently, a model might arrive at a correct answer via flawed reasoning, resulting in an unfaithful or logically inconsistent explanation. It may also internalize such flawed reasoning during training, which can hinder its ability to solve unseen tasks where the same shortcuts fail (Chua & Evans, 2025; Arcuschin et al., 2025; Chen et al., 2025).

To combat the lack of step-level supervision, prior works have proposed training strategies that provide feedback on intermediate reasoning steps (Lightman et al.; Uesato et al., 2022). While such finer-grained supervision has been shown to improve reasoning ability generally, it fails to evaluate the logical validity of the reasoning chains. For example, some approaches guide intermediate steps using token-level probabilities (Zhao et al., 2025; Li et al., 2025a) or learned reward models (Zhang et al., 2025b; Wang et al., 2024), but these feedbacks are inherently probabilistic and do not

---

*Email: jundong.xu@u.nus.edu
†Corresponding author: Hao Fei; Email: haofei37@nus.edu.sg

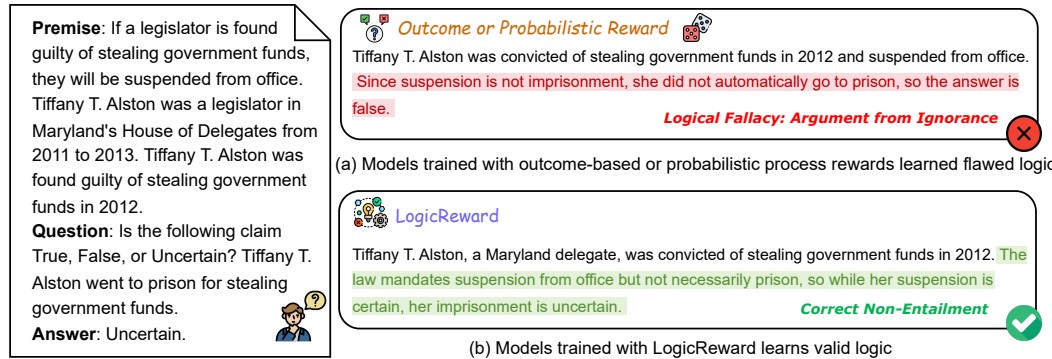

Figure 1: Comparison between models trained on data constructed with outcome-based or probabilistic process rewards and those trained with LogicReward.

provide a formal and symbolic assessment of logical validity. As a result, models trained in this way may produce reasoning that appears plausible but is logically flawed, as illustrated in Figure 1 (a). This limitation is particularly concerning in domains that demand strict logical guarantees, such as medicine and law. More fundamentally, evidence suggests that logical reasoning itself constitutes a critical driver of improved reasoning performance in large language models, playing a role analogous to large-scale supervision on code and mathematics (Xie et al., 2025; Liu et al., 2025b).

To bridge this gap, we propose **LogicReward**, a novel reward system that evaluates the formal logical validity of reasoning chains. LogicReward comprises two components: (1) **Autoformalization with Soft Unification** translates natural language into formal logic while capturing implicit assumptions and ambiguous information, thereby improving the formalization correctness; (2) **LogicScore** assesses reasoning quality by verifying the use of faithful information to avoid hallucination, and applying a theorem prover to check the logical validity. Using LogicReward, we construct logic-aware datasets and train LLMs via either supervised fine-tuning (SFT (Wei et al., 2022a)) or direct preference optimization (DPO (Rafailov et al., 2023)). Models trained using this data demonstrate stronger rigor and greater accuracy in reasoning, reducing illogical or unsound inferences.

We evaluate the model trained on datasets constructed with LogicReward on eight natural language inference and logical reasoning benchmarks. The model yields consistent improvements over strong baselines, with an 8B model surpassing the current state of the art by 2.0% with o4-mini (OpenAI, 2025b), 16.9% with DeepSeek-R1-8B (Guo et al., 2025), and 10% with GPT-4.1 (OpenAI, 2025a). Further analysis shows that the trained models outperform those trained with datasets derived from other leading reward systems. It also demonstrates greater generalizability on other reasoning tasks, such as math and commonsense, superior faithfulness, and effectiveness even in scenarios without ground-truth supervision. In summary, our main contributions are as follows:

- We propose **LogicReward**, a novel reward system for step-level reasoning's logical validity.
- Using LogicReward to construct datasets and following simple training procedures, we achieve new SoTA performance on Natural Language Inference (NLI) and Logical Reasoning tasks.
- The trained model shows strong generalizability to unseen tasks (e.g., mathematics, commonsense), along with improved faithfulness, and effectiveness in settings without ground-truth labels.
- We conduct extensive analyses of the efficacy of the proposed LogicReward mechanism, providing insights for future research.

## 2 RELATED WORK ON REASONING WITH LLMS

Early works improve reasoning through prompting strategies such as Chain-of-Thought (Wei et al., 2022b), Self-Consistency (Wang et al., 2023), Tree-of-Thoughts (Yao et al., 2023), and Least-to-Most (Zhou et al., 2023), which guide models to produce intermediate steps prior to arriving at the final answer. More recently, this approach has been scaled further. OpenAI o1 (Jaech et al., 2024) improves performance by allocating additional test-time compute and generating longer reasoning chains. Similarly, DeepSeek-R1 (Guo et al., 2025) employs reinforcement learning with

outcome correctness as the reward, thereby incentivizing longer reasoning chains. Building on these ideas, subsequent work has extended this approach to many different domains (Xie et al., 2025; Liu et al., 2025d; Jin et al., 2025). However, these methods rely primarily on outcome-level feedback without formally verifying the reasoning process, which can lead to explanations that are logically inconsistent (Chua & Evans, 2025; Arcuschin et al., 2025). In contrast, our proposed LogicReward introduces step-level logical evaluation, ensuring that reasoning is both correct and logically sound.

Prior work introduced process-level critics that evaluate intermediate reasoning steps rather than only final outcomes. Existing process-level critics can be broadly categorized into *probabilistic-based* and *symbolic-based* approaches. *Probabilistic methods* rely on models' internal signals to train models, such as entropy and token-level probabilities (Prabhudesai et al., 2025; Leang et al., 2025b; Fu et al., 2025), or process reward models (PRMs) trained with human or automatically generated step-level labels (Lightman et al.; Uesato et al., 2022; Zhang et al., 2025b; Wang et al., 2024), as well as using an LLM to assess step correctness (Zhang et al., 2025a; Gao et al., 2025; Xu et al., 2024). Although models trained on data constructed with such reward functions show improved reasoning coherence, they remain non-deterministic and often overlook subtle logical errors. Symbolic methods, in contrast, leverage external engines (e.g., theorem provers or code compilers) to verify reasoning deterministically. However, current work is mostly restricted to structured domains such as mathematics or programming (Jha et al., 2025; Leang et al., 2025a; Le et al., 2022; Gehring et al., 2025). In contrast, **LogicReward** applies formal logical evaluation to NLI, an inherently ambiguous domain that is harder to formalize but more reflective of everyday reasoning tasks.

Many other works use LLMs to tackle logical reasoning tasks by integrating symbolic structure or external solvers (Cheng et al., 2025; Liu et al., 2025a; 2023; Sun et al., 2024; Servantez et al., 2024; Liu et al., 2025c; Chen et al., 2026; Yang et al., 2026). For example, LAMBADA (Kazemi et al., 2023) applies backward chaining directly over natural language, while LINC (Olausson et al., 2023) and Logic-LM (Pan et al., 2023) translate natural-language statements into formal logic and rely on a prover to perform inference. SymbCoT (Xu et al., 2024) and Aristotle (Xu et al., 2025) extend this line by guiding LLMs to act as symbolic provers themselves. Other works incorporate prover feedback to iteratively refine or validate LLM-generated explanations (Quan et al., 2024a; 2025; 2024b). In contrast, LogicReward employs a theorem prover to construct informative rewards that guide LLM training, directly strengthening the model's internal logical-reasoning ability rather than relying on prompting or invoking a prover at inference time.

# 3 METHODOLOGY

In this section, we detail LogicReward, its role in data construction, and the training process.

## 3.1 DATASET COLLECTION AND ROLLOUT

We first sample around 6,000 instances, balanced across eight widely used NLI and logical reasoning datasets, including Multi-LogiEval (Patel et al., 2024), ProofWriter (Tafjord et al., 2021), FOLIO (Han et al., 2024), ProntoQA (Saparov & He, 2023), ProverQA (Qi et al., 2025), LogiQA (Liu et al., 2020), e-SNLI (Camburu et al., 2018), QASC (Khot et al., 2020). Further details are provided in Appendix C.1. Each instance consists of the premises $P = \{p_1, p_2, \ldots, p_m\}$ and a question $Q$. Given $(P, Q)$, we generate multiple candidate answers using Qwen3-8B (Yang et al., 2025) and GPT-4o (OpenAI, 2024), resulting in a total of **8** responses per question, denoted as $R = \{r_1, r_2, \ldots, r_n\}$. We denote all these generations as $D = \{R_1, R_2, \ldots, R_m\}$, where each $R_i$ contains the 8 responses to a single question. To ensure consistency in downstream reasoning analysis, we standardize responses $r$ by prompting the models to follow the format: Step 1: $s_1$; Step 2: $s_2$; ...; Step n: $s_m$; Answer: $A$. Therefore, each response consists of reasoning steps and a final answer, denoted as $r = \{s_1, s_2, \ldots, s_m\} \wedge A$.

## 3.2 LOGICREWARD

For effective reasoning, two aspects are crucial. First, the model must ground its reasoning in the given context, as using unsupported information undermines reliability. Second, the reasoning steps must also be logically valid; otherwise, the conclusions may be flawed. To capture these dimensions, we introduce **Premise Validity** and **Logic Validity**, corresponding to contextual grounding and

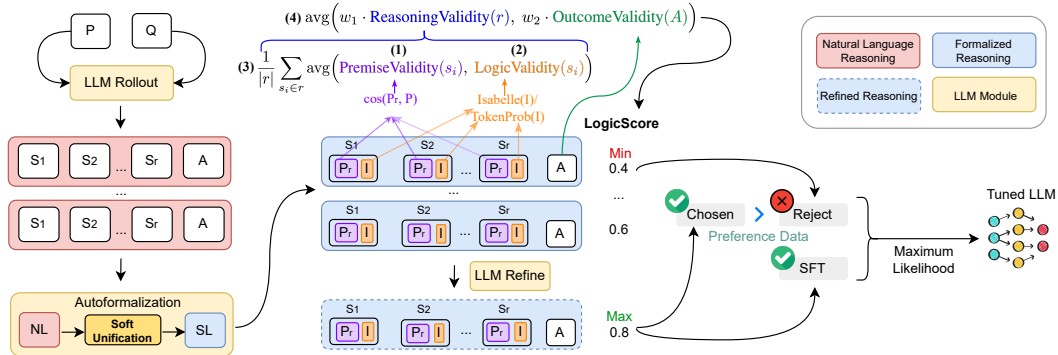

Figure 2: LogicReward Pipeline. We begin by rolling out responses with LLMs, followed by Autoformalization with Soft Unification. Each response is first assigned a Premise Validity score by comparing the used premises $P_r$ with the given premises $P$, after which Logic Validity checks inference $I$ using Isabelle. These are combined into Reasoning Validity, which is further integrated with Outcome Validity to yield the final LogicScore. The highest-LogicScore response is used to construct SFT data, while the responses with maximum and minimum LogicScores are paired to create DPO data for training.

logical soundness, respectively. For this purpose, we represent each reasoning step $s_i \in r$ as a pair $s_i = (P_r, I)$, where $P_r$ denotes the premise(s) used in the step, and $I$ represents the inference made based on $P_r$. We then define **Premise Validity** and **Logic Validity** in detail as follows.

**Premise validity.** For reliability, we measure Premise Validity by checking whether the referenced premises $P_r$ in each step are grounded in $P$. If $P_r$ contains multiple sentences, we split it as $P_r = \{q_1, q_2, \ldots, q_m\}$ where $m$ is the number of sentences. For each sentence $q_j$, we compute the cosine similarity with every given premise $p_k \in P$ as $c_{j,k} = \cos(q_j, p_k)$. We then set $c_j = \max_{p_k \in P} c_{j,k}$. Finally, the Premise Validity of step $s_i$ is defined as the average of these maxima:

$$\text{PremiseValidity}(s_i) = \frac{1}{m} \sum_{j=1}^{m} \max_{p_k \in P} \cos(q_j, p_k), \quad q_j \in P_r, \ p_k \in P \tag{1}$$

**Logic Validity.** Second, we evaluate Logic Validity to determine whether reasoning follows logical principles. To ensure logical guarantees, we use the theorem prover Isabelle (Paulson, 1994) to verify each inference step. However, reasoning needs to be converted into symbolic logic for the theorem prover to check, which is hard due to ambiguities and implicitness in natural language. For example, while humans recognize *Dad* and *Father* as equivalent, a symbolic system treats them as distinct predicates, causing validation errors. To address this, we design **Autoformalization with Soft Unification**. We prompt the LLM to supplement valid but implicit assumptions within each reasoning step that are not stated in $P_r$, thereby reducing ambiguity. This soft unification increases the likelihood of correct formalization and successful verification by the theorem prover. An illustrative example is shown in the Figure 3, and a more detailed analysis is provided in Section 5.5.

Following (Quan et al., 2024c), we parse the $P_r$ with Soft Unification into neo-davidsonia event semantics form (Parsons, 1990) and then convert to Isabelle/HOL theory (Nipkow et al., 2002) using an LLM. Isabelle will then evaluate their *syntactic* and *semantic validity*. Specifically, *Syntactic validity* checks whether the converted theory satisfies the grammar rules. *Semantic validity* checks whether the inference is logically valid, which can only be evaluated with correct syntax. If the syntax is invalid, we fall back to the confidence of the inference $I$, defined as the average token probability of the reasoning step: $\text{Conf}(I) = \frac{1}{|I|} \sum_{t \in I} \text{token\_prob}(t)$ for $I \in s_i$, where $I = (t_1, t_2, \ldots, t_k)$ denotes the inference of a reasoning step, and $t_j$ represents the $j$-th token in $I$. We then map them

into a unified **Logic Validity** score, and formally define it as follows:

$$\text{LogicValidity}(s_i) = \begin{cases} 1 & \text{if Syntax = True and Logic = True,} \\ 0 & \text{if Syntax = True and Logic = False,} \\ \text{Conf}(I \in s_i) \in [0,1] & \text{if Syntax = False.} \end{cases} \quad (2)$$

**Reasoning Validity.** To capture overall reasoning validity, we combine Premise Validity and Logic Validity, ensuring both faithfulness to the given context and the logical soundness of inference steps. We define the reasoning validity of a generation as the average, across all reasoning steps ($|r|$ in total), of the combined Premise Validity and Logic Validity at each step. Together, they form:

$$\text{ReasoningValidity}(r) = \frac{1}{|r|} \sum_{s_i \in r} \text{avg}\Big(\text{PremiseValidity}(s_i), \text{LogicValidity}(s_i)\Big), \quad \in [0,1] \quad (3)$$

**Outcome Validity.** While reasoning faithfulness is crucial, the final prediction must still match the ground truth. To capture this, we define an Outcome Validity, which takes the value 1 if the predicted answer matches the ground truth and 0 otherwise.

Finally, we integrate both reasoning validity and outcome validity as LogicScore, denoted as:

$$\textbf{LogicScore}(r, A) = \text{avg}\Big(w_1 \cdot \text{ReasoningValidity}(r), \ w_2 \cdot \text{OutcomeValidity}(A)\Big), \quad \in [0,1] \quad (4)$$

Here, $w_1$ and $w_2$ denote the weights assigned to reasoning validity and outcome validity, respectively. These weights satisfy $w_1, w_2 \geq 0$ and $w_1 + w_2 = 1$. For simplicity, we set $w_1 = w_2 = 0.5$, giving equal importance to reasoning faithfulness and final correctness. Each instance in $D$ is assigned a LogicScore, resulting in $D_r$.

### 3.3 REFINING REASONING WITH THEOREM PROVER FEEDBACK

Based on the above rewarding process, we observed that many LLM-generated reasoning steps were judged logically invalid, even when the final answer was correct. We then randomly sampled 300 instances for in-depth analysis and identified two main sources of error. We found that, first, some inferences are logically invalid, which is expected. Second, some other inferences appear valid but are judged invalid due to missing information, such as unstated assumptions, even after applying soft unification. To address these issues and improve data quality, we construct a synthetic subset, $D_{\text{refined}}$, which refines not only the inference but also the soft unification process. As shown in Figure 3, for a response judged logically invalid by Isabelle, we prompt the LLM to iteratively refine the **Soft Unification** in the reasoning step using the error messages, until the reasoning is judged logically valid or a maximum iteration limit is reached. For each question $Q$, we randomly pick 2 of its responses to refine. After refinement, each response is re-evaluated using the same procedure to obtain an updated LogicScore. Further analysis is provided in Section 5.5.

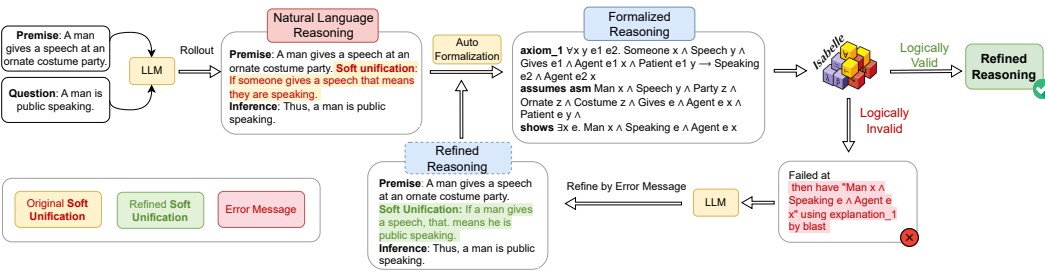

Figure 3: We present an illustrative example of leveraging Isabelle's error messages to refine the Soft Unification process. For simplicity, we show a single-step example.

| Model | M.-Logi. | PW. | FOLIO | Pron. QA | ProverQA | LogiQA | E-SNLI | QASC | Avg |
|---|---|---|---|---|---|---|---|---|---|
| *Non-Reasoning Models* | | | | | | | | | |
| Qwen2.5-7B | 63.0 | 52.7 | 53.9 | 95.3 | 63.3 | 59.0 | 69.7 | 97.3 | 66.2 |
| GPT-4o | 68.0 | 71.3 | 63.5 | 91.0 | 78.4 | 69.3 | 80.0 | 97.3 | 73.9 |
| GPT-4.1 | 67.7 | 79.0 | 68.8 | 95.3 | 76.3 | 67.3 | 78.7 | 97.3 | 75.0 |
| *Reasoning Models* | | | | | | | | | |
| Skywork-o1-8B | 50.9 | 33.3 | 33.3 | 37.8 | 44.3 | 38.0 | 31.7 | 72.9 | 41.2 |
| Nemotron-8B | 50.2 | 54.2 | 47.6 | 74.9 | 52.6 | 52.2 | 39.0 | 87.2 | 54.9 |
| MiniCPM-8B | 69.1 | 45.3 | 53.9 | 73.4 | 57.1 | 54.3 | 52.3 | 97.3 | 59.6 |
| Deepseek-R1-8B | 64.8 | 75.6 | 57.3 | 93.3 | 59.2 | 53.2 | 47.5 | 95.4 | 68.6 |
| QwQ-32B | 79.9 | 85.4 | 78.9 | 100.0 | 78.1 | 67.7 | 74.0 | 97.0 | 82.6 |
| o4-mini | 82.0 | 90.3 | 80.8 | 100.0 | 78.8 | 65.7 | 72.3 | 98.0 | 83.5 |
| *LogicReward* | | | | | | | | | |
| Llama3.1-8B-Instruct | 73.7 | 48.0 | 54.7 | 86.0 | 47.7 | 50.5 | 47.3 | 81.2 | 60.4 |
| **+LogicReward** | 79.0 | 48.4 | 57.1 | 90.3 | 60.1 | 56.0 | 62.1 | 97.8 | 71.4 |
| | (↑5.3) | (↑0.4) | (↑2.6) | (↑4.3) | (↑12.4) | (↑5.5) | (↑14.8) | (↑16.6) | (↑11.0) |
| Qwen3-8B | 82.5 | 94.5 | 62.8 | 100.0 | 87.2 | 67.9 | 63.5 | 91.3 | 82.3 |
| **+LogicReward** | 89.7 | 97.0 | 89.0 | 99.6 | 92.1 | 63.2 | 65.3 | 99.6 | 85.5 |
| | (↑7.2) | (↑2.5) | (↑26.2) | (↓0.4) | (↑4.9) | (↓4.7) | (↑1.8) | (↑8.3) | (↑3.2) |

Table 1: Main Results. Parentheses show change with LogicReward.

## 3.4 TRAINING

**Training Data Construction.** The final training set is constructed from $D_{\text{final}} = D_r \cup D_{\text{refined}}$. For training, we adopt SFT (Wei et al., 2022a) and DPO (Rafailov et al., 2023) due to their efficiency and lower computational cost. For SFT, we use the highest LogicScore for each $R \in D_{\text{final}}$ as the training target. For DPO, we select the response with the highest LogicScore as the *chosen* instance and the lowest as the *rejected* for each $R \in D_{\text{final}}$. Additional analyses for data constructing strategies and training algorithms are discussed in the Appendix F.2 and F.1.

**Training Procedure.** We use Llama3.1-8B and Qwen3-8B as base models and follow a two-stage training pipeline with SFT followed by DPO. We apply Low-Rank Adaptation (LoRA (Hu et al., 2022)) for computational efficiency. Further implementation details are provided in Appendix D.

## 4 EXPERIMENTS

### 4.1 EXPERIMENTS SETUP

**Dataset.** We evaluate on the test sets corresponding to the source dataset used for training. For each dataset, we randomly sample 300 questions whenever sufficient test data is available; otherwise, we use the entire test set. We measure accuracy by exact match.

**Baselines.** We include a broad range of baselines. These include non-reasoning models GPT-4o (OpenAI, 2024), GPT-4.1 (OpenAI, 2025a), and LLAMA3.1 (AI, 2024), and reasoning models post-trained for reasoning tasks, including O4-MINI (OpenAI, 2025b), DEEPSEEK-R1-8B (Guo et al., 2025), QWEN3-8B (Yang et al., 2025), MINICPM-8B (OpenBMB, 2024), NEMOTRON-8B (Nvidia, 2025), SKYWORK-O1 (AI, 2025), and QWQ-32B (Qwen Team, 2025).

Since these models may have been trained on different data sources using different base models, direct comparison is not entirely fair. To ensure fairness, we further establish baselines using the same backbone models (LLAMA3.1-8B), source data, and hyperparameters as LogicReward, but with alternative reward functions to construct training data from the source. Specifically, we consider (i) confidence-based reward (average token probability), (ii) LLM-as-Judge (GPT-4o (OpenAI, 2024)), and (iii) a process reward model based on NEMOTRON-70B-REWARD (NVIDIA, 2024).

**Settings.** For all non-reasoning models, we use temperature 0 to ensure reproducibility. For reasoning models, we follow recommended practices and set the temperature to 0.6. For O4-MINI, we use the default temperature, as no alternative setting is available. All models are under a zero-shot Chain-of-Thought setting. Details of the implementations are provided in Appendix D.

| Reward Systems | M.-Logi. | PW. | FOLIO | Pron. QA | ProverQA | LogiQA | E-SNLI | QASC | Avg |
|---|---|---|---|---|---|---|---|---|---|
| Confidence | 76.9 | 52.4 | 36.7 | 81.0 | 52.3 | 57.4 | 66.7 | 89.8 | 64.3 |
| LLM-as-judge | 65.8 | 40.3 | 51.5 | 84.6 | 47.5 | 51.5 | 62.7 | 95.3 | 60.2 |
| PRM | 66.7 | 59.1 | 52.4 | 90.6 | 62.1 | 54.3 | 56.4 | 97.0 | 66.0 |
| **LogicReward** | 79.0 | 48.4 | 57.1 | 90.3 | 60.1 | 56.0 | 62.1 | 97.8 | 71.4 |

Table 2: Comparison of the same model trained with different reward systems.

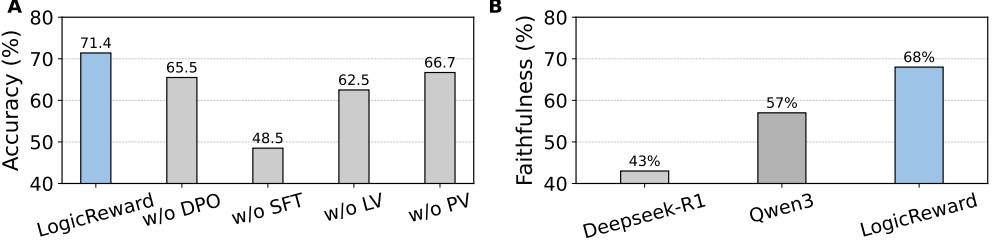

Figure 4: Panel **A** presents the ablation study. Panel **B** presents the faithfulness of different models.

## 4.2 MAIN RESULTS

**Overall, the model trained on data constructed by LogicReward achieves the best average performance.** As shown in Table 1, LOGICREWARD-QWEN surpasses strong baselines including DEEPSEEK-R1-8B, QwQ-32B, and O4-MINI by 16.9%, 2.9%, and 2.0%, respectively. Moreover, it attains SoTA or near SoTA results on most benchmarks. Furthermore, LOGICREWARD-LLAMA improves upon its base model by 11%, demonstrating the effectiveness of LogicReward across different architectures. These results demonstrate that LOGICREWARD substantially enhances reasoning robustness and establishes a new SoTA performance, without relying on scaling model size.

**Model trained on datasets constructed with LogicReward outperforms those trained on datasets derived from other reward functions.** To ensure a rigorous and fair evaluation of different reward methodologies, we conduct additional experiments where all models are trained under the same settings. As shown in Table 2, the model trained on datasets constructed with **LogicReward** achieves the highest overall average performance (71.4), surpassing Confidence, LLM-as-judge, and PRM by 7.1%, 11.2%, and 5.4%, respectively. We attribute these gains to the design of LogicReward, which is more closely aligned with the underlying reasoning objective. The *Premise Validity* component supervises models to ground their reasoning on correct premises, while the *Logic Validity* component ensures that inferences are logically valid. Training with this joint LogicReward helps the model not only select relevant evidence but also maintain rigorous logical consistency, leading to more robust reasoning performance.

## 5 ANALYSIS

We now delve deeper into our system to examine why it achieves such advances.

## 5.1 ABLATION STUDY

The ablation study in Figure 4A shows that the complete LogicReward-Llama model achieves the highest accuracy, confirming the necessity of integrating all components. Removing DPO reduces performance by 5.9%, highlighting the effectiveness of using LogicReward to construct preference data. The most severe degradation (22.9%) occurs when SFT is removed, demonstrating that LogicReward accurately captures reasoning quality; using the highest-LogicScore responses as SFT targets is therefore key to achieving this improvement. Excluding Logic Validity decreases accuracy by 8.9%, underscoring its importance in enforcing logical consistency and preventing reasoning shortcuts that compromise the answer's validity. Finally, removing Premise Validity results in a decline of 4.7%, indicating its contribution to teaching the model to use correct premises.

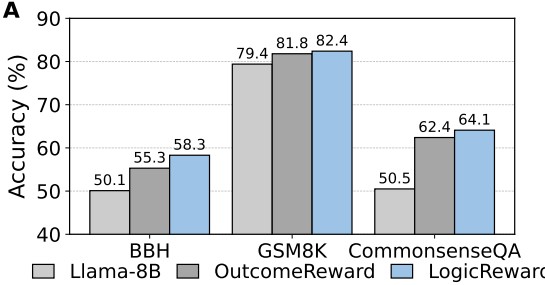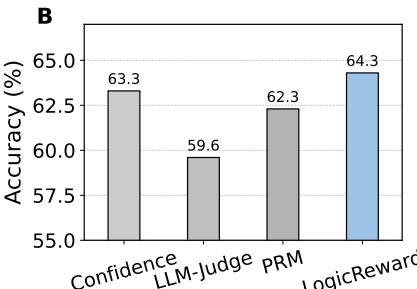

Figure 5: Panel **A** presents the performance of different models on previously unseen tasks. Panel **B** illustrates the performance when outcome information is excluded from the reward.

## 5.2 FAITHFULNESS

**LogicReward enhances reasoning faithfulness.** Since current training either uses outcome as the only reward without process supervision or probabilistic process signals, such as confidence, without logical guarantees, intermediate reasoning steps can still be logically flawed. This poses a critical problem in high-stakes domains such as medicine and law, where trustworthiness is paramount and even minor reasoning errors can lead to serious consequences.

Therefore, we evaluate the faithfulness of their outputs. We define unfaithfulness where the final answer is correct but produced through flawed reasoning, such as misusing premises, making invalid inferences, omitting critical steps, using shortcuts, or relying on unsupported assumptions.

To measure this, we randomly sampled 100 correct instances from models' responses and manually evaluated their reasoning using the strictest criteria. In Figure 4B, our results show that the **model trained on LogicReward-derived data achieves higher faithfulness, outperforming Deepseek-R1-8B and Qwen3-8B by 25% and 11%, respectively.** The improvements stem from LogicReward's design, in which reasoning must pass theorem-prover checks to earn a higher reward, providing rigorous, logic-verified supervision, which is stronger than outcome-only or other probabilistic process signals. This discourages shortcut behaviors (e.g., guessing the answer or skipping justifications) and promotes reasoning paths that are logically consistent with the premises. As a result, LogicReward helps produce more faithful reasoning traces, making them more trustworthy and interpretable in real-world applications.

## 5.3 GENERALIZABILITY TO UNSEEN TASKS

**LogicReward helps models improve generalizability.** To assess whether our model generalizes to unseen tasks, we evaluate on three out-of-distribution (OOD) benchmarks: BBH (Suzgun et al., 2023), GSM8K (Cobbe et al., 2021), and CommonsenseQA (Talmor et al., 2019). Compared to the base model and models that rely only on outcome-based reward to construct SFT and DPO training, LogicReward-Llama demonstrates stronger generalizability to OOD settings. As shown in Figure 5A, it achieves improvements of 8.2% on BBH, 3% on GSM8K, and 13.6% on CommonseQA. These results show that the method not only enhances in-domain performance but also transfers effectively to tasks not encountered during training.

The gains are most pronounced on BBH and CommonQA. For BBH, which demands structured and rigorous reasoning, LogicReward plays a key role by enforcing logical validity. For CommonsenseQA, the improvement can be attributed to our proposed **Soft Unification** strategy and the Synthetic Refined-Reasoning Dataset. Soft unification works by making implicit information and assumptions explicit, which aligns closely with the nature of commonsense reasoning, where humans often omit commonsense steps when reasoning. By explicitly surfacing these assumptions during training, the model learns to incorporate them into its reasoning process, thus improving its commonsense reasoning ability. The refined dataset further amplifies this effect by iteratively refining reasoning steps, including the Soft Unification, thereby improving the model's ability to incorporate commonsense knowledge during reasoning. In contrast, the gains on GSM8K are smaller, likely because mathematical reasoning relies less on commonsense knowledge and more on domain-

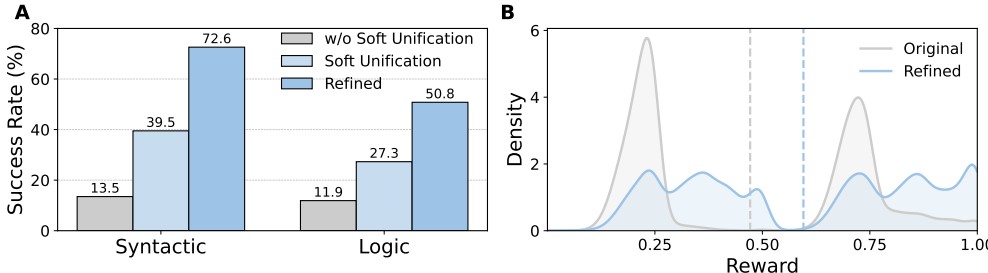

Figure 6: Panel **A** reports the success rates in terms of syntactic validity and logical validity, comparing reasoning without soft unification, with soft unification, and with refinement. Panel **B** presents the reward distributions for original reasoning and refined reasoning.

specific reasoning. Overall, these findings suggest that LogicReward enhances the model's ability to learn transferable reasoning principles, improving robustness and adaptability in OOD scenarios.

## 5.4 Learning Without Ground-Truth Outcome Supervision

**LogicReward outperforms other reward systems in the absence of ground-truth labels.** In many real-world applications, ground-truth answers are unavailable, necessitating the use of reward functions that do not require ground-truth supervision. To evaluate in this setting, we compare LogicReward against token probability, PRM, and LLM-as-judge, all without using outcome signals to construct the training data. As shown in Figure 5B, the model trained on data constructed with LogicReward achieves the highest average accuracy, surpassing Confidence, PRM, and LLM-as-judge by 1%, 2%, and 4.7%, respectively. These results underscore the advantage of LogicReward in label-free scenarios.

This is because token probability reflects a model's internal confidence but is purely statistical, offering no guarantee of logical correctness. PRMs, trained on step-by-step process labels, suffer from limited generalizability in domains lacking consistent fine-grained annotations. LLM-as-judge is also unreliable, as its evaluations can be inconsistent and subjective, driven by heuristics rather than robust logical criteria, making it an unstable reward signal. In contrast, **LogicReward** leverages theorem-prover signals that explicitly enforce logical validity. This design avoids reliance on statistical confidence or costly process labels, providing a principled, domain-agnostic reward signal for training reasoning models.

## 5.5 Analysis on Soft Unification and Reasoning Refinement

**Soft Unification improves syntactic correctness and Logic Validity.** As shown in Figure 6A, soft unification increases the rate of syntactic validity by improving syntactic correctness. This allows more reasoning traces to be verified by the Theorem Prover. Additionally, it enhances logical validity during theorem verification. The reason is that natural language often omits implicit information or assumptions that humans can easily interpret but symbolic systems cannot, leading the Theorem Prover to judge them as logically invalid even if they appear valid to humans. Soft unification leverages the LLMs' strong language interpretability to inject these assumptions into symbolic forms, thereby enabling the Theorem Prover to perform genuine verification.

**Our refinement strategy substantially improves reasoning quality.** As shown in Figure 6B, the grey two-humped pattern reflects the reward distribution before refinement. It shows a separation between incorrect and correct outcome rewards: the peak on the left corresponds to an outcome reward of 0, while the peak on the right corresponds to an outcome reward of 1. The main mass of the distribution lies in 0–0.25 on the left and 0.5–0.75 on the right. We further analyzed this pattern and found that it occurs because instances before refinement generally receive high premise validity, indicating that the model is often using correct premises; however, they also receive low logic validity, suggesting that much of the reasoning fails the theorem prover's logical check.

After applying our refinement method (blue distribution), we observe a more balanced spread across the entire 0–1 range, with notable increasing density in the 0.25–0.5 and 0.75–1 intervals. This shift indicates that the reward distribution is moving toward higher-quality reasoning. The improvement is further supported by the mean average reward, which rises from 0.47 to 0.6. Together, these results demonstrate that our refinement method effectively enhances the quality of reasoning.

## 6 CONCLUSION

This study introduces **LogicReward**, a novel reward system that leverages a theorem prover to capture logical correctness and guide model training. We further introduce *Autoformalization with Soft Unification*, which reduces natural language ambiguity and improves the formalization quality, enabling a more effective use of the theorem prover. With LogicReward, we construct supervised fine-tuning (SFT) datasets and preference datasets (e.g., for DPO) to train models. Experiments show that our method outperforms strong baselines such as o4-mini and GPT-4.1, demonstrating strong generalizability, faithfulness, and effectiveness even in the absence of ground-truth labels.

Future work could focus on two aspects. First, the auto-formalization process is still imperfect. Even with soft unification, certain ambiguities in natural language remain difficult to formalize. Future work could explore pipelines that proactively ask clarification questions to resolve these ambiguities, enabling more accurate auto-formalization. Second, the labeling process of LogicReward is relatively slower than approaches that only compare final outcomes. As a result, in on-policy reinforcement learning scenarios, training can take longer when using LogicReward. Future work should focus on optimizing the LogicReward pipeline for efficiency, allowing it to scale more effectively to training paradigms such as on-policy reinforcement learning.

## ACKNOWLEDGMENTS

This work is supported by the Ministry of Education, Singapore, under its MOE AcRF TIER 3 Grant (MOE-MOET32022-0001).

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

APPENDIX INDEX

This supplementary material includes the following sections:

## A    CLARIFICATION ON THE USE OF LARGE LANGUAGE MODELS

This work utilizes large language models as both research subjects and auxiliary tools throughout various stages of the research process. As research subjects, we employ Qwen3-8B and GPT-4o as the primary models for generating candidate reasoning responses in our dataset construction pipeline, where Qwen3-8B produces 5 candidate responses and GPT-4o produces 3 responses per question instance, resulting in 8 total responses per question across approximately 6,000 instances from natural language inference and logical reasoning datasets. These models serve as the foundation for our LogicReward system, where their generated reasoning chains undergo formal logical validity assessment through our Autoformalization with Soft Unification mechanism. Additionally, we utilize LLMs as integral components within our methodology, specifically for the autoformalization process that translates natural language reasoning into symbolic logic by parsing into neo-davidsonian event semantics form and converting to Isabelle/HOL theory format, as well as for supplementing valid but implicit assumptions within reasoning steps to reduce ambiguity during formal verification. Besides, we trained models Llama3.1-8B-Instruct and Qwen3-8B based on the constructed data. The baseline models in our evaluation framework represent state-of-the-art LLMs that serve as comparison benchmarks for assessing the effectiveness of our LogicReward-trained models. Beyond their role as research subjects and methodological components, LLMs have been employed as auxiliary tools for code development assistance and writing support throughout the research process. All LLM usage in this work adheres to the respective terms of service and ethical guidelines.

## B    ETHICS STATEMENT

This research was conducted in accordance with ethical guidelines for computational research and involves the development and evaluation of LogicReward, a novel reward system that evaluates the formal logical validity of reasoning chains in large language models. All datasets used in this study, including Multi-LogicEval, ProofWriter, FOLIO, ProntoQA, ProverQA, LogiQA, E-SNLI, QASC, BBH, GSM8K, and CommonsenseQA, are publicly available and widely used in natural language inference and logical reasoning research, with no new human subject data collected and all usage complying with original licensing terms and ethical guidelines. The study utilizes existing large language models (Qwen3-8B and GPT-4o) for candidate answer generation and evaluation, following respective terms of service and ethical guidelines provided by model developers. While this research aims to improve reasoning capabilities in AI systems for beneficial applications in education, scientific research, and decision support systems, we acknowledge the potential dual-use nature of advanced AI capabilities and encourage responsible deployment and further research into AI safety and alignment.

## C    Reprodicibility statement

To ensure reproducibility, we provide all necessary details and resources. We introduce the details of the LogicReward system in Section 3.2, with pseudocode in Appendix G and anonymized source code as supplementary material. We also report data construction procedures in Section 3.1 and Appendix C.1, baseline model settings in Appendix D, training parameters in Appendix E, and prompts in Appendix H. We will also make all code and data publicly available upon acceptance.

### C.1    Dataset Details

We provide details of each dataset used in our experiments. The training sets are used for model training, and the test sets for evaluation; for datasets without predefined splits, we perform our own random split.

**Multi-LogiEval** (Patel et al., 2024) is a comprehensive logical reasoning dataset with various inference rules and depths. It covers three logic types, propositional, first-order, and non-monotonic, consisting of more than 30 inference rules and more than 60 of their combinations.

**ProofWriter** (Tafjord et al., 2021) is a synthetic natural language reasoning over rule sets that pairs questions with formal proofs and abductive steps; used to test proof generation and QA at increasing depths.

**FOLIO** (Han et al., 2024) is a human-annotated, logically complex, and diverse dataset for reasoning in natural language (NL), equipped with first-order logic (FOL) annotations, sourced from real-world knowledge like Wikipedia.

**ProntoQA** (Saparov & He, 2023) is a synthetic QA built from first-order-logic "worlds," enabling formal parsing of chain-of-thought into symbolic proofs for analysis.

**ProverQA** (Qi et al., 2025) is created via ProverGen, which synergizes the generative strengths of Large Language Models (LLMs) with the rigor and precision of symbolic provers, enabling the creation of a scalable, diverse, and high-quality FOL reasoning dataset.

**LogiQA** (Liu et al., 2020) is a multiple-choice reading-comprehension question requiring deductive reasoning, sourced from Chinese civil-service exams and translated to English.

**e-SNLI** (Camburu et al., 2018) is a large-scale dataset for natural language inference (NLI). It contains about 570k sentence pairs (premise–hypothesis), each annotated with one of three labels: entailment, contradiction, or neutral.

**QASC** (Khot et al., 2020) is a multiple-choice science QA dataset requiring multi-hop reasoning by combining two supporting facts from a large corpus. It emphasizes compositional reasoning and retrieval.

We perform additional experiments using the same test set employed in the main study. In addition, to evaluate the out-of-distribution (OOD) generalization of our method, we introduce several datasets that are unseen during training and randomly sample 500 instances from each. Specifically, we include:

**BIG-Bench Hard** (Suzgun et al., 2023), a challenging benchmark designed to probe reasoning ability across diverse tasks. We use the BoarderGameQA specifically.

**GSM8K** (Cobbe et al., 2021), a collection of grade-school math word problems that require multi-step arithmetic reasoning.

**CommonsenseQA** (Talmor et al., 2019), which evaluates models' ability to perform commonsense reasoning over everyday scenarios.

## D    Baseline Details

We describe the details of our baselines and how we set them up in this section.

## D.1 Main Experiments Baselines

In Table 3, we introduce the detailed setting of the baseline models.

| Model | Model_ID | Temperature |
|---|---|---|
| GPT-4o (OpenAI, 2024) | gpt-4o-2024-08-06 | 0 |
| GPT-4.1 (OpenAI, 2025a) | gpt-4.1-2025-04-14 | 0 |
| LLaMA3.1 (AI, 2024) | meta-llama/Llama-3.1-8B-Instruct | 0 |
| o4-mini (OpenAI, 2025b) | o4-mini-2025-04-16 | default |
| DeepSeek-R1-8B (Guo et al., 2025) | deepseek-ai/DeepSeek-R1-Distill-Llama-8B | 0.6 |
| Qwen3-8B (Yang et al., 2025) | Qwen/Qwen3-8B | 0.6 |
| MiniCPM-8B (OpenBMB, 2024) | openbmb/MiniCPM4.1-8B | 0.6 |
| Nemotron-8B (Nvidia, 2025) | nvidia/Llama-3.1-Nemotron-Nano-8B-v1 | 0.6 |
| Skywork-o1 (AI, 2025) | Skywork/Skywork-o1-Open-Llama-3.1-8B | 0.6 |
| QwQ-32B (Qwen Team, 2025) | Qwen/QwQ-32B | 0.6 |

Table 3: Baseline models, their official IDs, and decoding temperatures.

## D.2 Strict Comparisons Baselines

To ensure strict and fair comparison, we compare different reward functions under the identical settings. For all baselines, we use Llama-3.1-8B-Instruct as our basemodel, and all baselines are trained with the same SFT plus DPO pipeline as LogicReward. For training data, we use the same source dataset $D$ explained in Section 3.1. We explain how we set up each baselines as follows:

**Token Probability.** For a response $r = \{s_1, s_2, \ldots, s_m\} \wedge A$, we compute the *average token probability* of each step $s_i$, denoted as

$$P(s_i) = \frac{1}{|s_i|} \sum_{t \in s_i} p(t),$$

where $p(t)$ is the model's probability for token $t$. The reasoning score is then obtained by averaging across all steps:

$$S(r) = \frac{1}{m} \sum_{i=1}^{m} P(s_i).$$

Finally, the overall reward also incorporates outcome correctness $\text{Acc}(A)$, giving the reward function:

$$R(r) = S(r) + \text{Acc}(A).$$

In Section 5.4, to evaluate the effectiveness of the reward functions in the absence of ground truth, we remove $\text{Acc}(A)$ and use $S(r)$ as the final reward.

we assign a reward to each response in $R = \{r_1, r_2, \ldots, r_n\}$. We first use the highest-rewarded response $r^+$ to construct the SFT dataset. We then form the DPO pair $(r^+, r^-)$ by selecting $r^+$ as the chosen instance and the lowest-rewarded response $r^-$ as the rejected instance.

**LLM-as-Judge.** For the rollout set $R = \{r_1, r_2, \ldots, r_n\}$, we provide the ground-truth answer to the LLM (GPT-4o with temperature 0) and prompt it to select the best response $r^+$ and the worst response $r^-$. For Section 5.4, we exclude that ground-truth answer from the prompt. We use $r^+$ to construct the SFT dataset, and the pair $(r^+, r^-)$ to construct the DPO dataset.

**Process Reward Model.** For each generation $r = \{s_1, s_2, \ldots, s_m\} \wedge A$, we use NEMOTRON-70B-REWARD to compute the reward for each step and take the average across all steps as the *reasoning reward*:

$$R_{\text{reason}}(r) = \frac{1}{m} \sum_{i=1}^{m} \text{Reward}(s_i).$$

We also compute an *outcome reward* based on the correctness of the final answer $A$, denoted as $R_{\text{outcome}}(A)$. Since these two rewards may differ in scale, we normalize both to the interval $[0, 1]$ using min–max normalization:

$$\hat{R}(x) = \frac{x - \min(x)}{\max(x) - \min(x)}.$$

Finally, the overall reward is defined as the average of the normalized reasoning and outcome rewards:

$$R_{\text{final}}(r) = \tfrac{1}{2}\left(\hat{R}(R_{\text{reason}}(r)) + \hat{R}(R_{\text{outcome}}(A))\right).$$

For Section 5.4, where ground-truth answers are unavailable, we exclude $R_{\text{outcome}}(A)$ and use only $R_{\text{reason}}(r)$ as the reward.

We then use the highest-rewarded response $r^+$ to construct the SFT dataset, and form the DPO dataset using the pair $(r^+, r^-)$, where $r^+$ is the chosen instance and $r^-$ is the rejected instance.

**Outcome Correctness.** In addition to the above baselines, we also evaluate a baseline that uses outcome correctness as the reward signal for dataset construction. Specifically, for SFT, we randomly select a rollout response $r \in R$ from $D$ such that $r$ contains the correct answer. For DPO, we randomly choose a correct response as the positive instance $r^+$ and an incorrect response as the negative instance $r^-$.

## E    TRAINING PATAMETER

For all training conducted in this paper, we adopt the Low-Rank Adaptation (LoRA (Hu et al., 2022) ) technique. All training was conducted using two H100 GPUs.

**SFT Training.** For all supervised fine-tuning (SFT), we use a batch size of 4 per device with gradient accumulation over 16 steps, yielding an effective batch size of 64. Training employs a linear learning-rate scheduler with an initial learning rate of $3 \times 10^{-5}$ and a warmup phase of 150 steps. To improve stability, we apply gradient clipping with a maximum norm of 0.3 and enable gradient checkpointing to reduce memory overhead. Optimization is performed with RMSprop, and trained with 1 epoch.

**DPO Training.** For direct preference optimization (DPO), training is conducted for 3 epochs with a batch size of 2 per device and gradient accumulation over 32 steps, giving an effective batch size of 64. The learning rate remains $3 \times 10^{-5}$, but we switch to a cosine scheduler for smoother convergence. We continue to use RMSprop and apply gradient checkpointing. To regulate policy updates, we set the DPO regularisation parameter $\beta = 0.1$, ensuring stability while maintaining alignment with human preferences.

## F    EXTENDED EXPERIMENTAL RESULTS

### F.1    COMPARISON OF DIFFERENT TRAINING PARADIGMS

We conduct additional experiments on the same test set as in the main experiments to compare different training paradigms, including SFT, SFT + DPO, and SFT + SimPO (Meng et al., 2024). For the baselines, we use outcome correctness as the sole reward to construct the SFT and DPO training sets.

As shown in Figure 7A, LogicReward consistently outperforms outcome-only supervision across all settings. Furthermore, beyond the experiments presented in the main results, we also evaluate SimPO as an alternative preference-learning paradigm. While SimPO provides further gains over SFT, the improvements are smaller than those achieved with DPO.

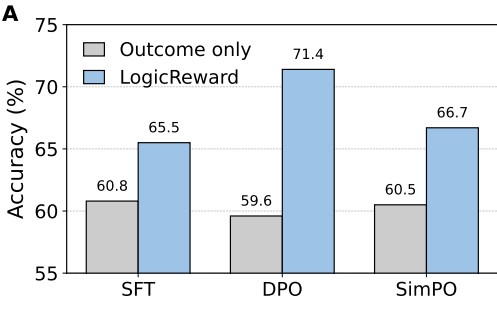 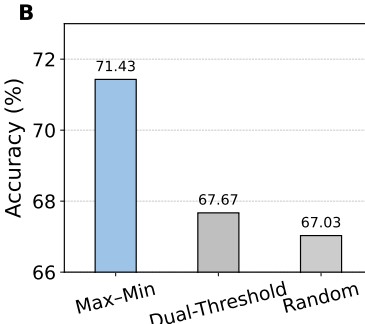

Figure 7: Panel **A** presents a comparison across training paradigms, while Panel **B** reports results for different data selection strategies.

### F.2 DIFFERENT DATA CONSTRUCTION STRATEGIES

We evaluate three strategies for constructing DPO pairs with responses scored by a reward model.

- **Top vs. Bottom (Max–Min).** For each question, we select the highest-rewarded response as the chosen sample $r^+$ and the lowest-rewarded response as the rejected sample $r^-$, yielding a single DPO pair $(r^+, r^-)$.
- **Dual-Threshold ($\geq 0.75$ vs. $\leq 0.25$).** For each question, all responses with reward $\geq 0.75$ are treated as chosen and all responses with reward $\leq 0.25$ are treated as rejected; this results in multiple DPO pairs for each question.
- **Random Above/Below 0.5 ($> 0.5$ vs. $< 0.5$).** For each question, we uniformly sample one response with reward $> 0.5$ as $r^+$ and one with reward $< 0.5$ as $r^-$ to form a single DPO pair.

As shown in Figure 7B, *Max–Min* yields the highest accuracy (71.43%), as selecting the globally best and worst responses maximizes the reward margin per pair and produces a strong learning signal. *Dual-Threshold* attains lower performance (67.67%): while it generates more pairs, the inclusion of near-threshold and redundant pairs dilutes pair quality and reduces the effective margin relative to Max–Min. *Random Above/Below 0.5* performs the worst (67.03%), since random sampling around a mid threshold introduces label noise (e.g., mediocre "chosen" responses) and typically yields small reward margins, limiting gains.

**Practical Takeaway.** When reward estimates are reliable, prefer *Max–Min* pairing for the strongest signal. Use *Dual-Threshold* to expand the preference dataset when broader coverage is desired, but expect smaller gains. Avoid *Random Above/Below* unless necessary, as its noisier signal and weaker margins hinder performance.

## G ALGORITHM

We introduce the full LogicReward algorithm, following its components of Premise Validity and Logic Validity.

---

**Algorithm 1** LogicReward

---

**Require:** Question $(P, Q)$ with premises $P = \{p_1, \ldots, p_m\}$ and question $Q$; a **Large Language Model**; rollout size $n=8$
1: Responses $\leftarrow$ Rollout(Large Language Model, $(P, Q), n$)
         // Generate $n=8$ formatted responses, each as a list of reasoning steps and a final answer.
2: **for all** response $\in$ Responses **do**
3:      sumPremise $\leftarrow 0$;    sumLogic $\leftarrow 0$
4:      **for all** step $\in$ response **do**
5:          sumPremise $\leftarrow$ sumPremise + PremiseValidity(step[premise], $P$)
6:          sumLogic $\leftarrow$ sumLogic + LogicValidity(step[inference], $P$)
         // PremiseValidity checks grounding in $P$; LogicValidity checks formal soundness.

7:     **end for**
8:     numSteps $\leftarrow \#\{$step $\in$ response$\}$
9:     **return** $\dfrac{\text{sumPremise} + \text{sumLogic}}{\text{numSteps}}$

                             // Average over steps gives the response-level Logic Score in $[0, 1]$.

10: **end for**

---

**Algorithm 2** Premise Validity

---

**Require:** Premises $P = \{p_1, \ldots, p_m\}$; a step text step containing the referenced premise(s)
1: similarityList $\leftarrow [\,]$
2: **for all** $s \in$ SplitByPeriod(step) **do**
3:    **for all** $p \in P$ **do**
4:        similarityList.Append$\big(\cos(s, p)\big)$

                     // Use cosine similarity between sentence $s$ and premise $p$ (embedding space).

5:    **end for**
6: **end for**
7: **return** max(similarityList)

                     // Highest match across all premises discourages hallucinated references.

---

**Algorithm 3** Logic Validity with Theorem Prover and Soft Unification

---

**Require:** Premises $P$; a step with one or more inferences; a **Theorem Prover**; a **LLM**; a **Prompt** auto-formalization with soft unification
1: **for all** inference $\in$ step **do**
2:    symbolic_logic $\leftarrow$ LLM(inference, soft unification **Prompt**)

                 // Transform the inference into symbolic logic with soft unification to reduce ambiguity.

3:    $(\text{syntaxOK}, \text{logicOK}) \leftarrow$ TheoremProver$(\widehat{\text{inference}})$
4:    **if** syntaxOK $=$ True **then**
5:        **return** 1 **if** logicOK $=$ True **else** 0

                   // Syntactically valid formulas receive binary semantic validation.

6:    **else**
7:        tokenValidity $\leftarrow$ AvgTokenProbability(Large Language Model, inference $\mid P$)
8:        **return** tokenValidity

                 // Fallback: average token probability in $[0, 1]$ when syntax is invalid.

9:    **end if**
10: **end for**

---

## H PROMPT

We use the following prompt to roll out responses:

> **Rollout Prompt**
>
> **System**
> You are a careful reasoner. Read the premise(s) and the question, then reason step-by-step using numbered steps.
>
> For **each step**, write exactly three lines in this order and wording: `Premise: Soft Unification: Conclusion:`
>
> **Formatting rules:**
> - Title each step as `Step N:`, where $N$ starts at 1 and increments by 1.
> - The `Premise` must be either (i) one of the given premises, or (ii) a `Conclusion` from a previous step.
> - The `Soft Unification` must be a commonsense, assumption, or implicit information that is contextually reasonable.
> - The `Conclusion` must be new information that logically follows from the `Premise` and the `Assumption`.

- After all steps, output a final line in the format: `Final answer:  [xxx].`
- If the question has choices (A), (B), (C), ... write *only* the option letter in the brackets (e.g., `[A]`).
- Otherwise, write only the required label (e.g., `[True]`, `[False]`, `[entailment]`, `[contradiction]`, `[neutral]`).
- Do not use XML tags. Do not add extra commentary before or after the steps or the final line.

**Example:**
Premise: Harry read a book. People who read books will be smart.
Question: Will Harry be smart? Answer with true/false/unknown.

**Reasoning:** Step 1: Premise: Harry read a book. People who read books will be smart. Assumption: Harry is a person. A person is people. Conclusion: Harry will be smart. Final answer: [True]

**Answer the question below**:
Premises: [PREMISES]
Question: [QUESTION]

