# OpenReview forum: "LogicReward: Incentivizing LLM Reasoning via Step-Wise Logical Supervision"
_ICLR.cc/2026/Conference — ICLR 2026 Poster_

### Official Review · Reviewer_L2rG · 2025-10-25

**Soundness:** 3
**Presentation:** 3
**Contribution:** 4
**Rating:** 8
**Confidence:** 4

**Summary:**

The paper proposes LogicReward: a step‑level reward for reasoning that averages (i) Premise Validity—a cosine‑similarity grounding check against the provided premises—and (ii) Logic Validity—a deterministic check with Isabelle/HOL. The combined LogicScore selects SFT targets and DPO pairs. On eight NLI/logic benchmarks, LogicReward yields notable gains and higher human‑audited faithfulness; it also transfers to BBH/GSM8K/CSQA and remains competitive without outcome labels.

**Strengths:**

* Substantial performance improvements across diverse NLI/logic datasets; the 8B backbones show consistent average gains (+11.0 for Llama3.1‑8B; +3.2 for Qwen3‑8B).
* Faithfulness improves and is human‑verified: on a manual audit of 100 correct cases, the LogicReward‑trained model achieves 68% faithful traces vs. 43%/57% for baselines. I think reasoning faithfulness is critical for transparency/trust in the era of thinking LLMs.

**Weaknesses:**

* The training data are constructed automatically (LLM rollouts + theorem‑provers), and the paper does not report end‑to‑end human auditing of that data. This raises the risk that intermediate steps/premises or injected “soft‑unification” assumptions contain errors. Though the final positive performance gain from the proposed method's training compensates for some.

**Questions:**

* Your method generates training data with an automated pipeline. Could errors/noise slip in, and where in the pipeline are they most likely (e.g., premise-grounding, auto-formalization/parsing, soft-unification, prover-guided refinement, fallback scoring)? Did you perform any human auditing, and at what scale?

---

> ### Author Response · Authors · 2025-11-17
> **Author response to reviewer L2rG**
>
> Thank you for your positive evaluation! We are glad to hear that you find the performance gains substantial and appreciate the improved faithfulness of the LLM under our method. We would like to answer your question below.
>
> ---
>
> **W1: Potential risk of steps/premises or "soft unification" contains errors**
>
> Thank you for raising this point. This is an very interesting question and we would like to clarify three aspects:
>
> Firstly, unlike purely model-generated supervision, LogicReward relies on symbolic validation from a prover, which discards all logically invalid steps. This provides the most rigorous safeguard than typical LLM-only training pipelines and significantly reduces the impact of noisy data. Thus, steps could not contain any logical error.
>
> Secondly, for the premise, this is captured by the PremiseValidity that calculates the cosine similarity of the used premise with the group truth premise. If the premise is wrong, it will get a low score, thus a penalty in training.
>
> Thirdly, regarding the risk of soft unification, we would like to clarify that incorrect assumptions generated by Soft Unification do not propagate unchecked. If an inferred assumption is invalid, the theorem prover simply rejects the entire reasoning step. The only potential risk is the hypothetical scenario where a step “passes” by generating an assumption that can pass the TP check, while it is unreasonable (i.e., a form of hacking by just repeating a target hypothesis). To examine this, we randomly sampled 100 prover-validated steps and manually inspected the Soft Unification assumptions. We found no suspicious or adversarial patterns, suggesting that such behavior might be very rare in practice.
>
> ---
>
> **Q1: error/noise slip in during data construction**
>
> We appreciate this important question. While we do not conduct large-scale human auditing, we have carefully examined where errors might occur and how our pipeline mitigates them.
>
> First, as illustrated in W1, we manually evaluated 100 randomly sampled steps and found no suspicious behaviour.
>
> Second, our empirical analyses show that the components we introduce—Soft Unification and refinement—reduce rather than introduce noise. As reported in Figure 6, Soft Unification increases syntactic success by 26% and logical validity by 15.4%; refinement further boosts these by 33.1% and 23.5%, and raises the average reward score by 13%. These gains indicate that the pipeline improves data quality by making more steps both formalizable and logically sound.
>
> Third, in stages verified by the theorem prover (e.g., autoformalization and refinement), errors cannot propagate because symbolic checking provides the strongest form of validation. Errors may arise only in stages not covered by the prover, such as premise grounding or fallback scoring you mentioned, which have a probabilistic nature. However, any inaccuracies in these components are still reflected in the LogicReward signal in a soft, probabilistic-based manner and are penalized during DPO training, preventing them from being reinforced.
>
> ---
>
> Thanks again for your positive feedback! We are happy to answer any further questions you may have.

---

### Official Review · Reviewer_x5nR · 2025-10-30

**Soundness:** 3
**Presentation:** 2
**Contribution:** 2
**Rating:** 6
**Confidence:** 3

**Summary:**

This paper introduces LogicReward, a novel reward system designed to train LLMs to produce reasoning that is not only correct in its final outcome but also "faithful" and "logically rigorous" at each intermediate step. The authors think that in existing training methods, rewarding only correct final answers can lead to models arriving at the right solution through "flawed reasoning", and supervising intermediate steps often relies on probabilistic signals, still "lacks guarantees of logical soundness".

To address these, the paper proposes LogicReward, which evaluates the logical validity of each reasoning step using a theorem prover. The method translates ambiguous natural language reasoning steps into the formal symbolic logic required by a theorem prover (specifically, Isabelle). Then, the paper gets the LogicScore by the following steps:
1. Checks if the reasoning step is "grounded in the given context". It measures the cosine similarity between the premises used in a step and the original premises provided in the problem.
2. Uses the theorem prover to check if the inference made in the step is logically sound. If the step cannot be parsed (i.e., has invalid syntax), the system falls back to using the "average token probability" as the score.
3. The average of the Premise Validity and Logic Validity scores across all steps in the reasoning chain.
4. A binary score (1 or 0) for whether the final answer matches the ground truth.
5. The final LogicScore is an equal-weighted average of the Reasoning Validity and the Outcome Validity.

After getting the LogicReward, the signal could be used to construct datasets and used to train models via SFT and DPO.

**Strengths:**

1. The primary strength is the integration of a symbolic theorem prover into the reward loop. This moves beyond the limitations of purely outcome-based or probabilistic process-based rewards, providing a more rigorous "formal and symbolic assessment of logical validity".
2. The method directly targets and improves reasoning faithfulness, a critical need for high-stakes domains like medicine and law. The paper's manual evaluation found that the LogicReward-trained model could produce more faithful reasoning processes.
3. The model shows strong generalization to out-of-distribution tasks, including BBH, GSM8K, and CommonsenseQA.
4. LogicReward excels in scenarios without ground-truth labels. By removing the "Outcome Validity" component, the reward signal relies only on Premise and Logic Validity. In this setting, it still outperformed other reward systems (LLM-as-judge, PRM, Confidence), demonstrating its value in real-world situations where labels are scarce.

**Weaknesses:**

1. The authors explicitly acknowledge that the LogicReward labeling process, which involves running a theorem prover, is "relatively slower than approaches that only compare final outcomes".
2. This speed limitation presents a significant challenge for scalability, particularly for "on-policy reinforcement learning scenarios"  (like PPO or GRPO) that require fast, in-the-loop reward generation. The paper's use of SFT and DPO  cleverly sidesteps this by being offline methods, but it leaves the system's viability for on-policy RL as an open question.
3. The auto-formalization process is "still imperfect". Even with soft unification and refinement, ambiguities in natural language remain difficult to formalize. The results show that even after refinement, the logic success rate was 50.8%, meaning nearly half of the steps were still not validated as logically sound by the prover.
4. When a reasoning step fails the syntactic check and cannot be evaluated by the theorem prover, the system "falls back to the confidence of the inference $I$, defined as the average token probability of the reasoning step". This pragmatic choice means that for a large portion of reasoning steps (those that fail to parse), LogicReward does not actually provide a symbolic logical guarantee.
5. The method was primarily trained and evaluated on NLI and logical reasoning datasets. While it showed some generalization, the paper notes that "gains on GSM8K are smaller". The authors suggest this is because mathematical reasoning relies "more on domain-specific reasoning"  than on the commonsense assumptions that Soft Unification is good at handling. This suggests the current pipeline may be less effective for highly specialized domains without further adaptation.

**Questions:**

Refer to Weaknesses.

---

> ### Author Response · Authors · 2025-11-17
> **Author response to reviewer x5nR**
>
> Thank you for your valuable feedback! We are glad that you recognize the importance and effectiveness of our methodology. We would now like to address the weaknesses you noted, as outlined below.
>
> ---
>
> **W1: Slower speed**
>
> We acknowledge that LogicReward involves a slower labeling process than evaluating only the outcome due to theorem proving, as also noted in the paper. However, this reflects a fundamental trade-off shared by any reward function that evaluates intermediate reasoning steps rather than only final outcomes. **Fine-grained, step-level supervision necessarily requires more computation**, but it produces significantly richer and more faithful reward signals—particularly important for long-horizon reasoning tasks, where outcome-only feedback is often insufficient and can lead to unreliable gradient signals. Importantly, this additional cost is incurred only during reward/data construction; model inference speed remains unchanged. We have clarified this point in the revision.
>
> ---
>
> **W2: generalizability on applying to on-policy scenarios**
>
> We thank the reviewer for highlighting the question of on-policy scalability. Our work intentionally adopts an off-policy training paradigm (SFT + DPO), which is substantially more sample-efficient and avoids the need for fast, in-the-loop reward computation. This choice reflects a practical consideration rather than a limitation of LogicReward itself. LogicReward can, in principle, be integrated with on-policy methods such as PPO or GRPO; however, doing so would require more computational resources and time, which is fundamentally tied to the trade-offs between off-policy and on-policy reinforcement learning.
> An exciting direction for future work is to improve LogicReward so that it can better support on-policy training—e.g., through faster reward evaluation, amortized reward models, or learned proxy reward functions that reduce the computational burden that on-policy RL inherently entails. We have added a discussion on this point in the revised manuscript.
>
> ---
>
> **W3 and W4: Imperfect formalization issue, then fall back to confidence**
>
> Auto-formalizing natural language has long been a fundamentally challenging problem. In this context, we view our Autoformalization with Soft Unification not as a limitation but as a meaningful advancement in addressing this difficulty.
>
> First, our method substantially improves the formalization success rate compared to prior approaches, demonstrating that soft unification effectively recovers many implicit or ambiguous elements that would otherwise cause failure, which offers a new perspective on optimizing autoformalization that the previous work does not have.
>
> Second, to the best of our knowledge, LogicReward is the first framework to use a theorem prover to reward ambiguous, free-form natural-language reasoning steps. Prior work has been largely limited to math or code, where reasoning is more formally structured and easier to verify. Our results show that theorem-proving-based rewards can extend to open-ended natural-language reasoning, demonstrating applicability beyond strictly formal domains and significantly improving the generalizability of this approach.
>
> Even though certain steps still fall back to confidence-based scoring when formalization is impossible, the ability to provide symbolic logical validation for a significant portion of steps already marks notable progress toward reliable, logic-grounded reward modeling, as illustrated in the experiments' results.
>
>
> ---
>
> **W5: Smaller gain on math dataset**
>
> We agree with the reviewer’s observation that LogicReward achieves relatively smaller gains on GSM8K compared to its improvements on NLI and deductive reasoning tasks. **Importantly, the “relatively smaller gain” is only in comparison to LogicReward’s larger improvements on other benchmarks, meaning that the performance increase on math is smaller relative to those tasks, not that LogicReward performs poorly in the math domain.** As shown in Figure 5A, on the GSM8K benchmark, **LogicReward still achieves stronger generalization than other baseline reward methods**, indicating that step-level, proof-guided supervision remains beneficial even in domains outside the core training distribution.
>
> This outcome is expected given the distributional differences: our training data primarily targets logical and NLI-style reasoning, whereas mathematical problem-solving in GSM8K requires domain-specific symbolic manipulation, algebraic transformation, and multi-step numerical reasoning—skills not explicitly covered in our training distribution. In contrast, NLI-style benchmarks align more naturally with the types of implicit commonsense assumptions that Soft Unification is designed to handle.
>
>
>
> ---
>
> Thanks for all the valuable feedback! We have made the corresponding revision in the manuscript and are happy to take any further questions you may have.

---

### Official Review · Reviewer_HUW3 · 2025-11-02

**Soundness:** 3
**Presentation:** 4
**Contribution:** 3
**Rating:** 6
**Confidence:** 3

**Summary:**

The paper targets the flaw of outcome-based training that can yield correct answers with incorrect reasoning, proposing LogicReward, which enforces step-level logical correctness via a theorem prover, plus Autoformalization with Soft Unification to reduce ambiguity and improve formalization. An 8B model trained on LogicReward data surpasses GPT-4o and o4-mini by 11.6% and 2% on natural language inference and logical reasoning, improves reasoning faithfulness, generalizes to math and commonsense, and provides a reliable reward signal without ground-truth labels. The authors plan to release data and code upon acceptance.

**Strengths:**

1. The use of symbolic methods for process supervision is well motivated and directly addresses limitations of outcome rewards. By enforcing step level logical validity, the approach reduces reward hacking and aligns training with faithful reasoning.
2. Incorporating theorem prover feedback in multiple iterations expands the exploration space and yields a more informative training signal. Iterative prover guided refinement improves coverage of candidate solutions, lowers variance, and encourages discovery of alternative valid derivations rather than overfitting to a single trace.
3. The manuscript is clearly written with well easy to follow and transparent evidence.

**Weaknesses:**

1. The paper omits several studies that combine LLMs with provers, for example Quan et al., “Verification and Refinement of Natural Language Explanations through LLM and Symbolic Theorem Proving,” EMNLP 2024. The task focus is closely related. Please situate LogicReward within this line of work and add direct comparisons where possible.
2. It is unclear how much the method’s gains depend on the chosen prover and logic. If a different proof assistant or formal language is used, does LogicReward retain its advantages and stability. Experiments that swap the prover or vary the logic and encoding would strengthen the generality claim.

**Questions:**

please refer to weaknesses

---

> ### Author Response · Authors · 2025-11-17
> **Author response to reviewer HUW3**
>
> Thank you for your valuable feedback. We appreciate your acknowledgement that our paper is well-motivated and clearly written. We would now like to address the weaknesses you have identified, as outlined below.
>
> ---
>
> **W1: Missing related work**
>
> Thank you for pointing out the relevant prior work. We would like to clarify that the mentioned EMNLP paper is already cited in our original manuscript (Line 190). Our framework is built upon this work, which functions as a component within LogicReward, so it is not appropriate for a direct comparison. To address your concern, we have added a dedicated paragraph in the related work section covering this line of research. If there are additional works we should include, we would be happy to add them.
>
> ---
>
> **W2: Generality Across Provers and Logics**
>
> Thank you for the thoughtful comment. We clarify that LogicReward is not tied to any specific prover or logic, though our chosen setting is intentionally selected as the most suitable for the task.
>
> **Prover Generality.**
> While our experiments use Isabelle/HOL, its Sledgehammer tool already calls multiple automated provers (e.g., E, Vampire, Z3). Therefore, LogicReward has effectively been evaluated across many different provers rather than relying on a single backend. The method itself is prover-agnostic and can be transferred to systems such as Lean; the main difference would be reduced efficiency due to the lack of Sledgehammer-style automation.
>
> **Logic Generality.**
> Our use of Neo-Davidsonian event semantics with First-Order Logic is motivated by the need to preserve as much semantic information as possible from natural language, following [1]. This reflects the design choice for maximally preserving semantic information, rather than a limitation of the method. LogicReward is compatible with other formalisms that offer similar semantic fidelity. We appreciate your suggestions and have clarified them in the revised manuscript.
>
> [1] Verification and Refinement of Natural Language Explanations through LLM-Symbolic Theorem Proving. EMNLP 2024
>
> ---
>
> We have made the corresponding revision, highlighted in red in the manuscript. Thank you again for your valuable feedback, and we are happy to address any further questions.

---

### Official Review · Reviewer_PCn6 · 2025-11-04

**Soundness:** 3
**Presentation:** 2
**Contribution:** 3
**Rating:** 6
**Confidence:** 3

**Summary:**

This paper introduces LogicReward, a reward system that enforces logical correctness of the reasoning traces of LLMs with a theorem prover. It also comes with "autoformalization with soft unification", which is a technique that prompts the LLM to provide implicit assumptions in the reasoning steps to reduce ambiguity in the statements. The resulting LogicScore is used to train LLMs under SFT and DPO settings. The overall method demonstrates solid performance increase on a variety of logical reasoning benchmarks, allowing small 8B models to outperform larger reasoning models. Furthermore, LogicReward performs better than several alternative reward systems such as LLM-as-judge.

**Strengths:**

- This paper tackles a good problem in LLM research, and the contribution is timely and generally principled.
- LogicReward achieves strong empirical performance. I appreciate that the authors evaluate the models/methods on a pretty large set of logical reasoning benchmarks.
- The study includes fair ablation studies and comparisons (e.g., Table 2 and Figure 4), corroborating the advantage of LogicReward.
- The generalizability analysis is interesting and encouraging (Sec. 5.3): LogicReward does not only help with strictly logical reasoning tasks—it may help with other kinds of reasoning too (e.g. grade school math).

**Weaknesses:**

- The related work section (Sec. 2) is incomplete. Much prior work has done on LLM logical reasoning using a range of different techniques, from earlier work like [1] to neurosymbolic approaches like [2, 3]. I think a dedicated paragraph is needed (could be in between "general reasoning" and "reward models") given the topic of this paper.
- While LogicReward seems a good fit for logical reasoning tasks, many kinds of reasoning are harder to (auto)formalize. Do the authors expect their method to work on harder mathematical problems like AIME or code generation tasks? It is not necessary to evaluate on other benchmarks here, but some more thorough discussion would be ideal. In other words, what are the soft and hard limitations of this approach?

[1] Kazemi, M., Kim, N., Bhatia, D., Xu, X., & Ramachandran, D. (2022). Lambada: Backward chaining for automated reasoning in natural language. arXiv preprint arXiv:2212.13894.
[1] Olausson, T. X., Gu, A., Lipkin, B., Zhang, C. E., Solar-Lezama, A., Tenenbaum, J. B., & Levy, R. (2023). LINC: A neurosymbolic approach for logical reasoning by combining language models with first-order logic provers. arXiv preprint arXiv:2310.15164.
[2] Pan, L., Albalak, A., Wang, X., & Wang, W. Y. (2023). Logic-lm: Empowering large language models with symbolic solvers for faithful logical reasoning. arXiv preprint arXiv:2305.12295.

**Questions:**

- Why specifically using the pipeline of SFT followed by DPO? A few more sentences of justifications/intuitions would be good.
- It would be good to clearly label or state how many test problems are used for each benchmark (wrt Sec 4.1).
- Minor: on Line 190, there is a typo. It should be "neo-davidsonia[n]".

---

> ### Author Response · Authors · 2025-11-17
> **Author response to reviewer PCn6**
>
> Thank you for your positive feedback of the importance of this work, our empirical results, and the analyses. We address the raised weaknesses and questions below.
>
> ---
>
> **W1: Related work section**
>
> Thank you for the suggestion. We have added a dedicated paragraph summarizing prior work on LLM-based logical reasoning, including all your cited works, in the revised Section 2.
>
> ---
>
> **W2: Applicability to harder math or code-generation tasks**
>
> Thank you for raising this question. We address it from three perspectives:
>
> Firstly, from a cognitive and computational perspective, logical reasoning is a core capability that underlies many other tasks. Strengthening this ability through LogicReward benefits not only logical inference but also downstream tasks such as mathematics and programming, which both depend heavily on structured, multi-step reasoning. Our experiments also show that reasoning gains on logical tasks transfer beyond the training domain.
>
> Secondly, the LogicReward framework is solver-agnostic. For more advanced mathematical tasks, the backend prover can be replaced with Lean; for coding tasks, it can be replaced with a code interpreter or execution-based verifier. This requires only swapping the solver component rather than changing the methodology itself.
>
> Thirdly, we agree with your point that many tasks are hard to formalize. But we want to offer a new perspective to look at it.
> A key challenge in autoformalization, observed in our work, is handling ambiguous or implicit information that natural language omits. This motivated our Autoformalization with Soft Unification module, which explicitly extracts hidden assumptions to produce faithful logical forms, which previous work does not improve autoformalization from this perspective. We believe similar issues arise in mathematics (e.g., unstated lemmas or implicit domain constraints) and in programming (e.g., unspecified input conditions or edge cases). The ideology behind soft unification in LogicReward, recovering implicit structure before verification, extends naturally to these domains, though task-specific adaptations would be required.
>
> In summary, we believe that the core principles of LogicReward can be extended to more complex mathematical and coding tasks, provided that they are adapted to align with the respective domain-specific solvers and formalization requirements. The primary challenge, of course, lies in autoformalization. Addressing this will likely require the development of soft unification techniques capable of recovering the rich, implicit information embedded within these more sophisticated mathematical and programming problem settings.
>
> Hope this answers your question!
>
> ---
>
> **Q1: Why SFT followed by DPO?**
>
> We adopt the SFT followed by the DPO pipeline because the two stages serve complementary roles. SFT provides the model with basic competence by imitating high-quality reasoning traces, which stabilizes subsequent optimization. DPO then refines the model by learning preference distinctions between better and worse solutions. This practice is consistent with prior alignment literature, where preference optimization (e.g., RLHF or DPO) is shown to be most effective when initialized from an SFT model rather than from a base model [1, 2].
>
> [1] Training language models to follow instructions with human feedback. NeurIPS 2022.
>
> [2] Direct Preference Optimization: Your Language Model is Secretly a Reward Model. NeurIPS 2023.
>
> ---
>
> **Q2: Number of test problems**
>
> Thank you for pointing this out. We have included the detailed number in the revision.
>
> ---
>
> **Q3: Typo on Line 190**
>
> Thank you for catching this. We have corrected "neo-davidsonia" to “neo-Davidsonian.”
>
> ---
>
> We have made reivision correspondingly in the manscript highlighted in red. Thank you again for your helpful comments. We appreciate the opportunity to improve the paper and are happy to address any further questions.

---

### Author Response · Authors · 2025-11-17
**General response to all reviewers**

Dear Reviewers,

Thank you for your time and valuable feedback, which has greatly helped us improve the quality of our work. We are encouraged by the positive overall assessment. Reviewers highlighted that the paper is timely and well-motivated (Reviewer PCn6, HUW3), demonstrates strong empirical gains (Reviewer PCn6, x5nR, L2rG), shows promising generalization (Reviewer PCn6, x5nR), includes solid analysis and ablations (Reviewer PCn6, x5nR), and is clearly written (Reviewer HUW3). We sincerely appreciate these encouraging remarks.

We have carefully considered all comments and provided detailed responses to each point. In addition, we have revised the manuscript accordingly, including:

1. Adding a Related Work section covering recent research papers on LLMs for logical reasoning;

2. Clarifying the role and usage of the theorem prover;

3. Improving the discussion and motivation for the SFT+DPO training pipeline; and

4. Expanding the Future Directions in conclusion for LogicReward.

We truly appreciate your constructive feedback and hope that our responses address your concerns. If there are any remaining questions, we would be very happy to clarify them.

Thank you again for your time and thoughtful feedback.

---

### Author Response · Authors · 2025-11-27

Dear Reviewers,

As we approach the end of the discussion period next week, we would like to check whether there are any remaining points you would like us to address or elaborate on. Please let us know if any additional clarification would be helpful. Your insights have been highly valuable throughout this process.

Thank you for your time and consideration.

---

### Meta-Review · Area_Chair_PbYc · 2026-01-07

**Summary:**

- **PCn6** (6): Incomplete related work on LLM logical reasoning; applicability to harder math/code tasks; justification for SFT+DPO pipeline; missing test set size details
- **HUW3** (6): Omitted relevant work (Quan et al., EMNLP 2024); questions about generality across different provers/logics
- **x5nR** (6): Slow theorem-prover labeling limits scalability; imperfect formalization (50.8% success rate); fallback to token probability undermines symbolic validation; smaller gains on GSM8K
- **L2rG** (8): Absence of end-to-end human audit of automatically constructed training data; risk of undetected errors in pipeline components

**Reviewer Concerns:**

Almost all concerns raised by the reviewers are addressed.

**Reviewer Scores:**

No reviewer expressed the will to change the score but all scores are positive already.

---

### Decision · Program_Chairs · 2026-01-26

Accept (Poster)